# In Vitro Virucidal Activity of Different Essential Oils against Bovine Viral Diarrhea Virus Used as Surrogate of Human Hepatitis C Virus

**DOI:** 10.3390/antibiotics13060514

**Published:** 2024-05-31

**Authors:** Gianvito Lanave, Francesco Pellegrini, Francesco Triggiano, Osvalda De Giglio, Maria Stella Lucente, Georgia Diakoudi, Cristiana Catella, Arturo Gentile, Roberta Tardugno, Giuseppe Fracchiolla, Vito Martella, Michele Camero

**Affiliations:** 1Department of Veterinary Medicine, University of Bari Aldo Moro, 70010 Valenzano, Italy; gianvito.lanave@uniba.it (G.L.); francesco.pellegrini@uniba.it (F.P.); mariastella.lucente@uniba.it (M.S.L.); georgia.diakoudi@uniba.it (G.D.); cristiana.catella@uniba.it (C.C.); arturo.gentile@uniba.it (A.G.); vito.martella@uniba.it (V.M.); 2Interdisciplinary Department of Medicine, Hygiene Section, University of Bari Aldo Moro, 70124 Bari, Italy; francesco.triggiano@uniba.it (F.T.); osvalda.degiglio@uniba.it (O.D.G.); 3Department of Pharmacy-Pharmaceutical Sciences, University of Bari Aldo Moro, 70125 Bari, Italy; roberta.tardugno@uniba.it (R.T.); giuseppe.fracchiolla@uniba.it (G.F.)

**Keywords:** bovine viral diarrhea virus, human hepatitis C, essential oils, *Salvia officinalis*, *Melissa officinalis*, citrus lemon, *Rosmarinus officinalis*, *Thymus vulgaris*

## Abstract

The hepatitis C virus (HCV) is a major hepatotropic virus that affects humans with increased risk of developing hepatocellular carcinoma. The bovine viral diarrhea virus (BVDV) causes abortion, calf mortality and poor reproductive performance in cattle. Due the difficulties of in vitro cultivation for HCV, BVDV has been used as surrogate for in vitro assessment of the efficacy of antivirals. Essential oils (EOs) display antiviral and virucidal activity on several viral pathogens. In this study, the virucidal activity of five EOs, *Salvia officinalis* L. EO (SEO), *Melissa officinalis* L. EO (MEO), *Citrus lemon* EO (LEO), *Rosmarinus officinalis* L. EO (REO) and *Thymus vulgaris* L. EO (TEO) against BVDV was evaluated in vitro at different concentrations for several time contacts. MEO and LEO were able to considerably inactivate BVDV with a time- and dose-dependent fashion. MEO and LEO at the highest concentrations decreased viral titer by 2.00 and 2.25 log_10_ TCID_50_/50 μL at 8 h contact time, respectively. SEO, REO and TEO displayed mild virucidal activity at the highest concentrations for 8 h contact times. In this study, the virucidal efficacies of MEO and LEO against BVDV were observed regardless of compound concentration and contact time. Further studies are needed to confirm the potential use of MEO and LEO as surface disinfectants.

## 1. Introduction

Flaviviruses are small (~50 nm) enveloped spherical viruses that possess a single genomic RNA of positive-sense polarity encoding three structural and seven non-structural proteins [1]. Flaviviruses belong to the family *Flaviviridae* which includes four distinct genera: *Hepacivirus*, *Orthoflavivirus*, *Pegivirus* and *Pestivirus* according to the Internation Committee on Taxonomy of Viruses (ICTV) [2]. Flaviviruses infect mammals and birds, and many are host-specific and pathogenic, such as the hepatitis C virus (HCV) in the genus *Hepacivirus*. Most members of the genus *Orthoflavivirus* are arthropod-borne, and many are important human and veterinary pathogens (e.g., yellow fever virus, dengue virus and West Nile virus) [2]. Pegiviruses have been identified in a variety of mammalian hosts, with transmission of human pegiviruses occurring by sexual, parenteral and maternal routes. Pestiviruses infect pigs and ruminants, including cattle, sheep, goats and wild ruminants [2].

The genus *Pestivirus* comprise 11 different species (from A to K) identified from different hosts. Bovine pestiviruses are members of the species Pestivirus A (bovine viral diarrhea virus 1, BVDV-1), B (BVDV-2) and H (Hobi-like pestivirus, HoBiPeV). Bovine viral diarrhea (BVD) is an infectious disease causing serious economic losses in cattle breeding worldwide [3].

The genus *Hepacivirus* includes HCV, a pathogen able to cause an infection associated with high risk of development of liver cirrhosis and hepatocellular carcinoma in humans [4]. Annually, HCV causes 400,000 deaths, and in the US, deaths from HCV have currently exceeded those induced by the human immunodeficiency virus (HIV) [5]. HCV treatment mainly include a combination of two or more antiviral drugs, displaying different mechanisms of action and an efficacy greater than 95% [6,7,8]. Combinations of more antivirals prevent the development of drug resistance and increase antiviral response [9]. 

The identification of an HCV-like virus, the rat hepacivirus, has proposed a novel small animal model for vaccine testing. However, the constraint of the model is due to the limited sequence homology with the HCV [10,11]. HCV-like hepaciviruses have been identified in other animal species including horses, dogs and wildlife species [12], although their pathobiology is still unknown.

BVDV and HCV share structural organization and genomic features. Although they possess a different genome size, both viruses possess a large polyprotein encoding structural (envelope and capsid proteins) and non-structural proteins. Moreover, BVDV and HCV share common steps of viral replication, being able to attach to the same low-density lipoprotein (LDL) receptor of the host cells [13]. Based on physicochemical similarities, BVDV has been widely used as a surrogate for HCV [14,15,16]. 

Chemical disinfectants, such as chlorine-based compounds, hydrogen peroxide and quaternary ammonium compounds, have been used to inactivate members of the family *Flaviviridae* by targeting the envelope or RNA genome [17]. Moreover, thermal treatment and exposition to UV rays of HCV induced direct damage of the viral genome [17]. Among flaviviruses, BVDV is regarded as one of the most challenging enveloped viruses to be inactivated [18].

Essential oils (EOs) are natural compounds extracted from various plants and contain a complex mixture of bioactive molecules with several unique biological and therapeutic properties [19]. Among their components, EOs display terpenes, terpenoids, carotenoids, coumarins and curcumin which are involved in anti-microbial, anti-inflammatory, antioxidant and other beneficial properties [20]. The use of EOs in different areas (i.e., medicine, microbiology, agriculture, food production, etc.) has gained growing interest also for their limited environmental impact [21,22]. EOs have shown antiviral activity against several pathogenic viruses [23] due to their ability to interfere with viral infections [24,25]. Moreover, EOs are able to affect the integrity of the viral envelope and capsid, thus inactivating viruses [26,27].

*Melissa officinalis* L. EO (MEO) displayed in vitro antiviral efficacy against severe acute respiratory syndrome coronavirus 2 (SARS-CoV-2), the herpes simplex virus (HSV) and the human immunodeficiency virus (HIV) through various mechanisms [28]. *Citrus lemon* EO (LEO) demonstrated virucidal activity against feline calicivirus used as a norovirus surrogate [26]. *Thymus vulgaris* L. EO (TEO) has already proven to be effective against several RNA viruses including CoVs of human and animal origin [25,29,30]. *Rosmarinus officinalis* L. (REO) has shown antiviral activity against herpesviruses [31] (Al-Megrin et al., 2020) for which also *Salvia officinalis* L. EO (SEO) displayed efficacy [32]. SEO exhibited antiviral efficacy also against avian influenza H5N1 virus and a moderate virucidal activity against BVDV used as a surrogate model for HCV [33].

The current study is an effort carried out to assess the in vitro virucidal activity of five commercial EOs (MEO, LEO, REO, SEO and TEO) against BVDV-1 used as surrogate for HCV. Outbreaks of HCV in health-care settings have been frequently described due to contaminated medications or equipment and violation of aseptic standard procedures [34,35,36]. The potential efficacy of EOs as disinfectants against cell-free viruses could provide a safe alternative tool at low environmental impact to reduce the risk of contamination of HCV on different high-touch surfaces of hospital settings.

## 2. Results

### 2.1. Analytical Details of EOs

The results of the gas chromatographic analysis hyphenated with mass spectrophotometry (GC/MS) of the tested EOs are shown in Table 1, Table 2, Table 3, Table 4 and Table 5. About seventeen components of SEO were identified and accounted for 98.03% of the total detected constituents (Table 1). The major components were eucalyptol (29%), camphor (23%), α-pinene (9.5%), camphene (8.8%), β-pinene (8.4%), lynalin acetate (5%), β-linalool (3.8%), endo-borneol (2.5%), α-terpineol (2%) and bornyl acetate (2%) (Table 1).

Twenty-seven compounds were identified in MEO, accounting for 91.71% of the total (Table 2). MEO was characterized by the presence of citral (43%), caryophyllene (25%), humulene (4.4%), limonene (4.3%), caryophyllene oxide (2.2%), geraniol (2%) and geranyl acetate (2%) (Table 2). 

The analysis of LEO revealed a complex mixture mainly consisting of oxygenated and hydrocarbon monoterpenes (Table 3). There were twenty different components accounting for 87.8% of the mixture. The six major detected compounds were limonene (53%), β-pinene (14.5%), γ-terpinene (5.9%), citral (3.8%), α-pinene (2.4%) and β-thujene (2%) (Table 3).

About twenty-five components of REO were characterized and accounted for 98% of the total detected constituents (Table 4). The major components were Eucalyptol (23%), α-pinene (20%), camphor (20%), camphene (8%), β-pinene (4%), endo-borneol (4%), o-cymene (3.5%), α-terpineol (2.5%), β-myrcene (2.2%) and caryophyllene (2.1%) (Table 4).

Twenty-five components of TEO were identified accounting for 98.35% of the total detected constituents (Table 5). The major components were thymol (47%), p-cymene (20%), γ-terpinene (9%), β-linalool (4%), α-pinene (2%), camphene (2%) and caryophyllene (2%) (Table 5). 

### 2.2. Cytotoxicity

The cytotoxicity of EOs at different concentrations for 72 h was assessed through evaluation by microscopy. The magnitude and diversity of cellular structural changes were dose-dependent [37]. Figure 1 displays the cytotoxic effects of MEO, LEO, TEO, REO and SEO at the highest concentrations (8670 μg/mL, 8360 μg/mL, 8900 μg/mL, 8660 μg/mL and 8790 μg/mL, respectively) compared to the untreated Madin Darby bovine kidney (MDBK) cells regarded as the control (Figure 1A–F).

The cytotoxicity of EOs was also assessed though the evaluation of cell viability by the XTT colorimetric by measuring the absorbance signal spectrophotometrically. The maximum noncytotoxic concentration was evaluated as the concentration at which the viability of the treated Madin Darby bovine kidney (MDBK) cells decreased to 20% compared to the control cells (CC_20_). In all the experiments, DMSO showed no effect on the cells. The untreated cells were used in each experiment as the negative control and considered as 0% cytotoxicity. The cytotoxicity of EOs on the MDBK cells, expressed as percentages, was plotted against the compound concentrations (Figure 2). 

The cytotoxicity of the MDBK cells treated with MEO at the higher concentrations (8670, 4335, 2167.5 and 1083.75 μg/mL) ranged from 61.5 to 69.25% and decreased from 54.5 to 0.82% at the lower concentrations (541.87, 270.93, 135.46, 67.73 and 33.71 μg/mL) (Figure 2A). Cytotoxicity of the MDBK cells treated with LEO at the higher concentrations (8360, 4180, 2090, 1045 μg/mL) ranged from 64.8 to 72.45% and decreased from 53.25 to 22.7% at the lowest concentrations (522.50, 261.25, 130.63, 65.31 and 32.66 μg/mL) (Figure 2B). 

The cytotoxicity of the MDBK cells treated with TEO at the higher concentrations (8900, 4450, 2225 and 1112.5 μg/mL) ranged from 58.95 to 68.4% and decreased from 36.1 to 0.5% at the lower concentrations (556.25, 278.12, 139.06, 69.53 and 34.76 μg/mL) (Figure 2C). 

The cytotoxicity of the MDBK cells treated with REO at the higher concentrations (8660, 4330, 2165 and 1082.5 μg/mL) ranged from 49.1 to 64.75% and decreased from 38.75 to 0.45% at the lower concentrations (541.25, 270.62, 135.31, 67.65 and 33.82 μg/mL) (Figure 2D). 

The cytotoxicity of the MDBK cells treated with SEO at the higher concentrations (8790, 4395, 2197.5 μg/mL) ranged from 39.05 to 58.75% and decreased from 20.0 to 0.5% at the lower concentrations (1098.75, 549.37, 274.68, 137.34, 68.67 and 34.33 μg/mL) (Figure 2E). 

Based on the adjusted dose–response curves, the CC_20_ value of the EOs were calculated. The CC_20_ for LEO was equal to 29.86 μg/mL, MEO accounted for 123.86 μg/mL, whilst TEO displayed a CC_20_ equal to 197.78 μg/mL. REO exhibited a CC_20_ equal to 346.40 μg/mL, and the CC_20_ of SEO was equivalent to 1098.75 μg/mL.

### 2.3. Virucidal Activity

BVDV was pretreated with EOs at concentrations 10- and 100-fold above the cytotoxic threshold for different contact times (10 and 30 min and 1, 4 and 8 h) and compared to the respective virus control (Figure 3 and Figure 4). SEO, only when used at a concentration 100-fold over CC_20_ (109,875.00 μg/mL), displayed significant virucidal effect after 8 h of contact, decreasing the viral titer by 0.75 log_10_ Tissue culture infectious dose (TCID_50_/50 μL) (*p* = 0.0127) as compared to control virus (Figure 4). MEO at a concentration 10-fold above the cytotoxic threshold (1238.57 μg/mL) was able to significantly reduce the viral titer by 0.75 (*p* = 0.0086) and 1.00 log_10_ TCID_50_/50 μL (*p* = 0.0015) after 4 and 8 h of contact, respectively, when compared with the control virus (Figure 3). MEO at a concentration 100-fold over CC_20_ (12,385.70 μg/mL) showed a significant virucidal effect against BVDV reducing the viral titer by 0.75 log_10_ TCID_50_/50 μL starting from 10 min (*p* = 0.0015) up to 1 h (*p* = 0.0127), by 1.25 log_10_ TCID_50_/50 μL after 4 h (*p* = 0.0002) and by 2.00 log_10_ TCID_50_/50 μL after 8 h (*p* < 0.0001), as compared to the virus control (Figure 4). LEO at a concentration 10-fold above the cytotoxic threshold (298.60 μg/mL) significantly reduced the viral titer by 0.75 log_10_ TCID_50_/50 μL, starting from 1 h (*p* = 0.0127) up to 8 h (*p* = 0.0086), as compared to the control virus (Figure 3). LEO at a concentration 100-fold over CC_20_ (2986.00 μg/mL) displayed significant virucidal effect against BVDV at all the contact times. The compound was able to decrease the viral titer by 1.25 log_10_ TCID_50_/50 μL at 10 min (*p* = 0.0002), 2.00 log_10_ TCID_50_/50 μL at 30 min and 1 h (*p* < 0.0001) and 2.25 log_10_ TCID_50_/50 μL at 4 h and 8 h (*p* < 0.0001), as compared with the virus control (Figure 4). REO, only when used at a concentration 100-fold over CC_20_ (34,640.00 μg/mL) after 8 h of contact, showed significant virucidal activity against BVDV decreasing the viral titer by 0.75 log_10_ TCID_50_/50 μL (*p* = 0.0127) (Figure 4). TEO, only when used at a concentration 100-fold over CC_20_ (19,777.80 μg/mL) was able to significantly reduce the viral titer by 1.00 (*p* = 0.0015) and 1.50 (*p* < 0.0001) log_10_ TCID_50_/50 μL after 4 and 8 h of contact, respectively (Figure 4).

## 3. Discussion

Alternative preparations of chemical sanitizers based on natural and plant sources have been explored to exclude toxicity problems. Several plant extracts have been reported with broad-spectrum antiviral properties [38]. Herbal preparations have been regarded safe for the environment and more tolerable for human health with respect to the antiviral pharmacological therapy, thus allowing us to expand the range of therapeutic tools against viral diseases [39]. Disinfectants based on natural resources have been tested effective against viral pathogens with comparable results observed with alcoholic-based formulations [40].

EO-based disinfectants have been proposed for replication inhibition or inactivation of the influenza virus and coronavirus in vitro [41,42,43]. The benefits of using EOs as alternative disinfectants comprise the application on porous surfaces that cannot be successfully achieved by conventional chemical disinfectants as well as the potential to integrate EOs in blends able to counteract microbes at different replication stages [44].

In this study, the virucidal activity of five commercially available EOs against BVDV was assessed. Despite being regarded as a severe pathogen for the cattle industry, BVDV has been used as a surrogate for HCV for the evaluation of virucidal effects of antiviral agents [15]. In certain features of viral replication, the use of BVDV is more beneficial than the HCV replicon system [45,46] in which early (attachment, entry and uncoating) or late steps (virion assembly and release) of the viral replication cycle cannot be examined. Despite more robust HCV cell culture systems having been subsequently reported [47], assessing the antiviral efficacy of some compounds against pestiviruses may provide relevant information for the design of novel antiviral strategies against HCV. Antiviral agents targeting host enzymes involved in BVDV replication would likely inhibit the replication of HCV [48].

In this study, the virucidal activity of SEO, MEO, LEO, REO and TEO was tested at concentrations 10- and 100-fold exceeding the cytotoxic threshold to evaluate its possible application as a surface disinfectant. The rationale of testing EOs at higher concentration relies on their potential use as virucide, since the compound is not posed into immediate contact with the cells. Although moderate cytotoxic effects were observed in the wells inoculated with the undiluted EO–BVDV mixture and in those inoculated with the 10^−1^ and 10^−2^ dilutions, this did not hamper the ability to verify the possible presence of CPE.

In this study, SEO displayed significant virucidal activity, reducing the BVDV titer by 0.75 log_10_ TCID_50_/50 μL only when applied at the at a concentration 100-fold over CC_20_ (109,875.00 μg/mL) after 8 h contact. In a previous report, SEO showed mild in vitro virucidal activity against BVDV after 2 h of contact [33]. Conversely, Schnitzler et al., observed the in vitro virucidal efficacy of SEO against herpes simplex virus type 1 (HSV-1) and 2 (HSV-2), reducing plaque formation by 90% for HSV-1 and by >99% for HSV-2 [32]. 

MEO and LEO, tested in our experiments 10-fold and 100-fold concentrated over the cytotoxic limit, displayed at both concentrations mild to moderate virucidal activity against BVDV. Both EOs, when applied at a concentration 100-fold above CC_20_ (MEO at 2986.00 μg/mL and LEO at 12,385.70 μg/mL), were able to significantly decrease viral titers starting from 10 min contact in a dose- and time-dependent fashion. At the longest time contact (8 h), MEO and LEO reduced the BVDV titer by up to 2.00 and 2.25 log_10_ TCID_50_/50 μL, respectively. 

In a previous report, MEO, when used at a concentration 1.5 μg/mL, considerably lower than those of MEO used in the present study, demonstrated in vitro virucidal activity against HSV-1, affecting its attachment to host cells [49]. MEO also displayed antiviral effects against other enveloped viruses through different mechanisms, i.e., preventing HSV-1 from binding to host cells, hindering HSV-1 replication post-adsorption, hampering the activity of the main protease and spike protein of severe acute respiratory syndrome coronavirus type 2 (SARS-CoV-2) or inhibiting avian influenza virus through the different virus replication stages [28,50]. 

In a former study, LEO displayed significant inhibitory effects in epithelial cells versus the angiotensin-converting enzyme 2 (ACE2) receptor, a host cell receptor, that play a crucial role in SARS-CoV-2 cell entry [51]. Moreover, in another study, LEO used at 3020.00 μg/mL induced a moderate in vitro virucidal effect against the non-enveloped feline calicivirus reducing viral titer as high as 1.25 log_10_ TCID_50_/50 μL after 8 h of time contact [26]. Despite virucidal efficacy of EOs against both several enveloped RNA and DNA viruses having been demonstrated [52], the activity of EOs against non-enveloped viruses requires further confirmation. In a recent study [53]), the same EOs assessed in the present study have been tested against a non-enveloped virus thus observing less relevant results. Accordingly, EOs are not regarded as a valid alternative for inactivation of non-enveloped viruses [54].

In our study, GC/MS revealed that MEO and LEO shared ten components (α-pinene, β-thujene, β-pinene, limonene, β-linalool, citral, geranyl aceate, caryophyllene, α-bergamotene, caryophyllene oxide), albeit being present with different percentages in the respective EO. Identifying the main active component in EOs used in our study would help to reduce the cytotoxic effect of each compound mixture. The major abundant component of EOs is often able to exert the most effective antiviral effect. However, less represented components of EOs should be also investigated for their antiviral activity. For instance, the bicyclic sesquiterpene caryophyllene was identified as the main active components of MEO displaying virucidal activity against HSV-1 and HSV-2 comparable to that of pure MEO, although without cytotoxic effects [55]. The monoterpenes β-pinene and limonene exhibited high anti-HSV-1 activity by direct interaction with free virus particles reducing viral infectivity by 100% [56]. Moreover, citral reduced the in vitro infectivity of the yellow fever virus when the virus was treated before and after entry into the host cells [57].

In our experiments, REO showed a significant mild virucidal effect after 8 h contact with BVDV only when concentrated 100-fold over CC_20_. Antecedent in vitro studies revealed that REO at 30 µg/mL caused 55% inhibition of HSV-1 plaques, whereas REO at 40 µg/mL caused 65% inhibition of HSV-2 plaques. Accordingly, rosemary extract has been suggested as a topical prophylactic or therapeutic agent for herpes viral infections [31]. REO is an aromatic plant of the family *Lamiaceae* whose main components include camphor, 1,8-cineole, α- and β-pinene and camphene [58]. In a previous report, camphene derivatives showed broad antiviral activities in vitro against a panel of enveloped pathogenic viruses, including the influenza virus, the Ebola virus (EBOV), and the Hantaan virus [59].

In this study, TEO, only when used at a 100-fold concentration over the cytotoxic threshold (19,777.80 μg/mL), was able to decrease the BVDV titer up to 1.50 log_10_ TCID_50_/50 μL after 8 h contact. In another report, TEO hindered in vitro replication of another enveloped virus, the feline coronavirus (FCoV), inducing significant reduction in the viral titer by 2 log_10_ TCID_50_/50 μL. Moreover, TEO at a concentration of 270 μg/mL, markedly lower than those of TEO used in the present study, determined a reduction in FCoV titer as high as 3.25 log_10_ TCID_50_/50 μL up to 1 h of time contact [25]. TEO has demonstrated antiviral activity also against other enveloped viruses, i.e., HSV1 and HSV-2 [60], the Newcastle disease virus [61], the influenza virus [62], the porcine reproductive and respiratory syndrome virus [63] and the infectious bronchitis virus [29]. 

The comparison of the results obtained in this study with other reports is hindered by different chemical compositions of EOs that depend on the harvest time of the plant, method of preparation, storage conditions and expiration of the compounds; moreover, some features of the plant, i.e., (the age of the plant, meteorological conditions during their growth, soil composition, light, temperature and level of rainfall) could also affect the parallelism of this study with previous studies. These variations limit the possibility of standardizing the protocols used for EOs. Moreover, the experiments performed to assess antiviral activity of EOs rely on different materials and methods.

## 4. Materials and Methods

### 4.1. Essential Oils

SEO, MEO, LEO, REO and TEO were provided by Specchiasol Srl, (Bussolengo, VR, Italy) and conserved at +4 °C until use. Solvents (analytical grade), a standard mixture of C10–C40 n-alkanes, and all standard compounds were acquired from Supelco Sigma-Aldrich Srl (Milan, Italy). The filters were furnished by Agilent Technologies Italia Spa (Milan, Italy). MEO, REO and SEO were extracted from the leaves of the respective plants. To obtain TEO and LEO, the flowering plant and the fresh peel were used, respectively.

### 4.2. GC/MS

All the EO solutions 1:100 *v*/*v* in ethyl acetate were analyzed by the GC-MS technique using an Agilent 6890N GC coupled with a 5973N MSD HP ChemStation. The gas chromatograph is equipped with a HP-5 MS capillary column. The time–temperature program was set to 5 min at 40 °C followed by a temperature ramp of 4 °C/min to 280 °C. For the identification of EO components, comparisons were made with the authentic standards available in the authors’ laboratory whenever possible. For each compound, the linear retention index (LRI) was calculated using a homologous series of C10-C40 alkane standard mixture [64]. The arithmetic index (AI) and similarity index/mass spectrum (SI/MS) were derived from a reference text [65] and/or the NIST 2017 Database, respectively [NIST 17, 2017. Mass Spectral Library (NIST/EPA/NIH). Gaithersburg, USA: National Institute of Standards and Technology. Last access 22 April 2024], and were determined as previously reported [66,67].

### 4.3. Cells and Virus

The MDBK cells were utilized and grown in an incubator at 37 °C and 5% CO_2_ in a humidified atmosphere. Dulbecco’s Modified Eagle Medium (D-MEM) containing 10% fetal bovine serum, 100 IU/mL penicillin, streptomycin 0.1 mg/mL, and 2 mM l-glutamine was used for the cells. The CP BVDV-1 strain NADL was grown, titrated in MDBK cells and displayed a titer of 10^3.50^ TCID_50_/50 μL. The stock virus was conserved at −80 °C until use.

### 4.4. Cytotoxicity Assay

The in vitro Toxicology Assay Kit XTT (Sigma–Aldrich Srl, Milan, Italy), based on 3-(4,5-dimethylthiazol-2 yl)-2,5-diphenyl tetrazolium bromide, was applied to assess the cytotoxicity of the EOs. The assay was carried out as formerly reported [30]. To evaluate the cytotoxicity of the EOs at different concentrations, 24 h confluent monolayers of MDBK cells grown in 96-well plates were used. The EOs were initially diluted in 1 mL of 0.1% dimethyl sulfoxide (DMSO; Sigma-Aldrich, St. Louis, MO, USA) and subsequently in D-MEM. SEO was determined at 8790, 4395, 2197.5, 1098.75, 549.37, 274.68, 137.34, 68.67, 34.33 μg/mL, MEO was evaluated at 8670, 4335, 2167.5, 1083.75, 541.87, 270.93, 135.46, 67.73, 33.71 μg/mL, LEO was assessed at 8360, 4180, 2090, 1045, 522.50, 261.25, 130.63, 65.31, 32.66 μg/mL, REO was analyzed at 8660, 4330, 2165, 1082.5, 541.25, 270.62, 135.31, 67.65, 33.82 μg/mL, and TEO was tested at 8900, 4450, 2225, 1112.5, 556.25, 278.12, 139.06, 69.53, 34.76 μg/mL.

For all the assays, untreated cells and cells treated with equivalent dilutions of DMSO without EOs were arranged as the control and vehicle control, respectively. The percentage of cytotoxicity for each EO was estimated though the following formula: % cytotoxicity = [(OD control cells − OD treated cells) × 100]/OD control cells.

The maximum noncytotoxic concentration was assessed and regarded as CC_20_. The experiments were performed in triplicate.

### 4.5. Virucidal Activity Assay

A total of 100 μL of undiluted BVDV was treated with different EOs (900 μL) at room temperature for several interval times (10 min, 30 min, 1 h, 4 h and 8 h). The virucidal effects against BVDV of SEO, MEO, LEO, REO and TEO at concentrations 10-fold (10,987.00 μg/mL, 1238.57 μg/mL, 298.60 μg/mL, 3464.00 μg/mL and 1977.78 μg/mL, respectively, and 100-fold (109,870.00 μg/mL, 12,385.70 μg/mL, 2986.00 μg/mL, 34,640.00 μg/mL and 19,777.80 μg/mL, respectively) over the cytotoxic threshold, regarded as CC_20_ were evaluated. The different virus–EO mixtures were titrated on the MDBK cells and compared with the untreated infected cells (control virus, CV), as previously described [53]. All the experiments were carried out in triplicate.

### 4.6. Viral Titration

Ten-fold dilutions (10^−0^–10^−8^) of each supernatant were titrated in quadruplicate in 96-well plates containing MDBK cells. The plates were conserved in an incubator at 37 °C and 5% CO_2_ humidified atmosphere for 72 h. Based on cytopathic effect (CPE), the viral titer was estimated relying on the Reed and Muench method [68].

### 4.7. Data Analysis

The data calculated by GC/MS and virucidal activity were summarized as area % ± standard error of mean and mean ± SD, respectively. EO concentrations were adjusted in log_10_, and the results obtained by the cytotoxicity assay were estimated by a non-linear curve fitting. A dose–response curve was developed through non-linear regression analysis to estimate the goodness of fit. From the fitted dose–response curves calculated from each assay, CC_20_ was evaluated. 

The normality of the data (Shapiro–Wilk test) and the homogeneity of variances (Levene median test) were evaluated. If both conditions were satisfied, the effects of different time contact for each compound concentration were analyzed by One-way ANOVA or a corresponding non-parametric test, followed by the Dunnet test as a post hoc test. The statistical significance level was always set at 0.05. The statistical analyses were carried out by the GraphPad Prism v8.1.2 program (Intuitive Software for Science, San Diego, CA, USA).

## 5. Conclusions

In conclusion, we described the in vitro virucidal activity of five EOs using BVDV as a surrogate of HCV. MEO and LEO were able to decrease considerably BVDV infectivity in our experimental design. Further investigations are required to improve the performance of both EOs, to investigate the mode of action on BVDV and to determine if other BVDV strains might display different susceptibility to MEO and TEO.

These results provide alternative perspectives/tools for the sanitization of surfaces and objects potentially contaminated by hepaciviruses, although indirect transmission of HCV is not regarded as a major risk in humans. Also, potential applications could be conceived as soon as the role of hepaciviruses in domestic animals is understood.

## Figures and Tables

**Figure 1 antibiotics-13-00514-f001:**
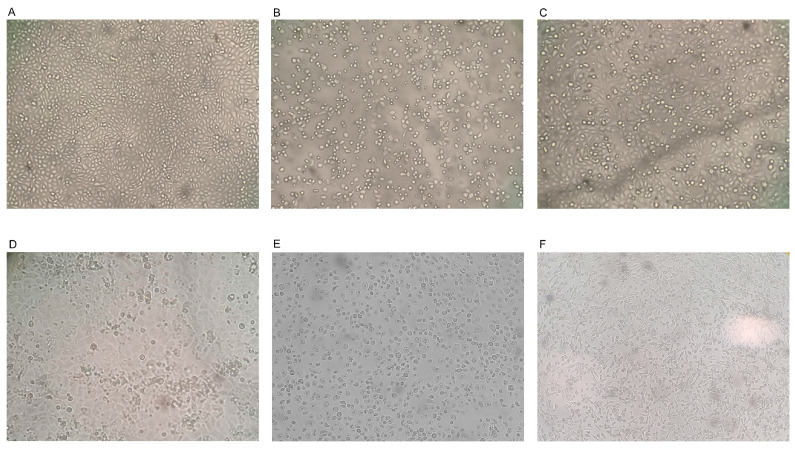
Live-cell imaging (magnification 10×) of Madin Darby bovine kidney (MDBK) cells. Untreated (control) MDBK cells (**A**) and those treated with *Melissa officinalis* L. essential oil (MEO) (**B**), *Citrus lemon* EO (LEO) (**C**), *Thymus vulgaris* L. EO (TEO) (**D**), *Rosmarinus officinalis* L. EO (REO) (**E**) and *Salvia officinalis* L. EO (SEO) (**F**).

**Figure 2 antibiotics-13-00514-f002:**
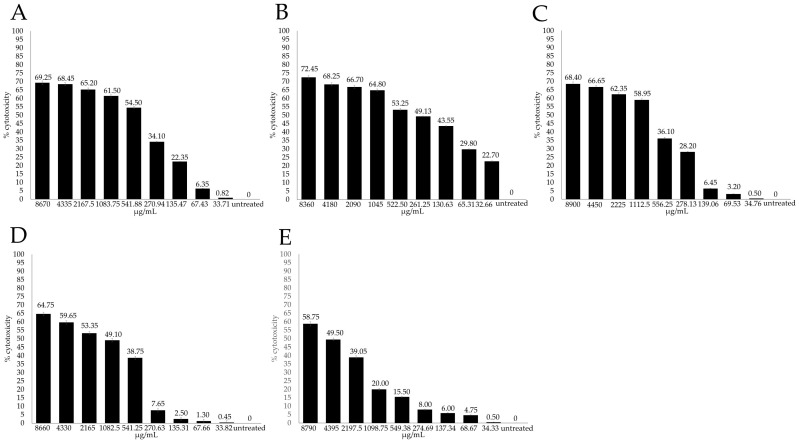
Cytotoxicity of the MDBK cells treated with *Melissa officinalis* L. essential oil (MEO) (**A**), *Citrus lemon* EO (LEO) (**B**), *Thymus vulgaris* L. EO (TEO) (**C**), *Rosmarinus officinalis* L. EO (REO) (**D**) and *Salvia officinalis* L. EO (SEO) (**E**) at 72 h post-treatment and calculated by the XTT assay. The value was calculated by setting 0% as the percentage of cytotoxicity in the untreated controls. Cytotoxicity is plotted against different concentrations (μg/mL) of EOs. Bars in the figures indicate the means. Error bars indicate the standard deviation.

**Figure 3 antibiotics-13-00514-f003:**
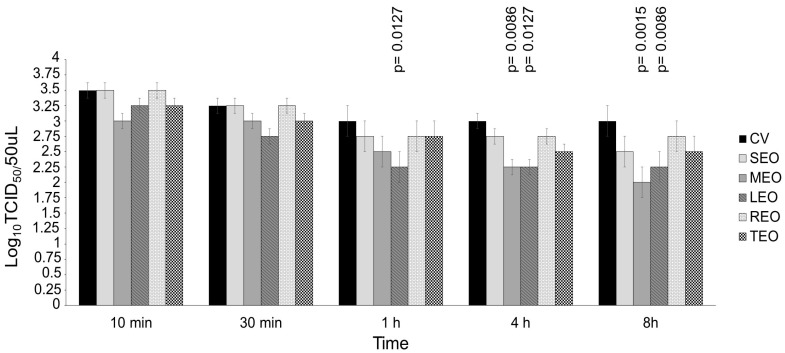
Virucidal activity of *Salvia officinalis* L. essential oil (SEO), *Melissa officinalis* L. essential oil (MEO), *Citrus lemon* essential oil (LEO), *Rosmarinus officinalis* L. essential oil and *Thymus vulgaris* L. essential oil at concentrations 10-fold over the cytotoxic threshold against the bovine viral diarrhea virus (BVDV) for 10 min, 30 min, 1 h, 4 h and 8 h at room temperature. The different virus–EO mixtures were titrated on Madin Darby bovine kidney (MDBK) cells and compared with the untreated infected cells (control virus, CV). SEO was used at 10,987.00 μg/mL, MEO was used at 1238.57 μg/mL, LEO was used at 298.60 μg/mL, REO was used at 3464.00 μg/mL and TEO was used at 1977.78 against BVDV. The viral titers of BVDV were expressed as log_10_ TCID_50_/50 μL and plotted against the EOs at different concentrations and CVs. Significant *p* values are displayed. Bars in the figures indicate the means. Error bars indicate the standard deviation.

**Figure 4 antibiotics-13-00514-f004:**
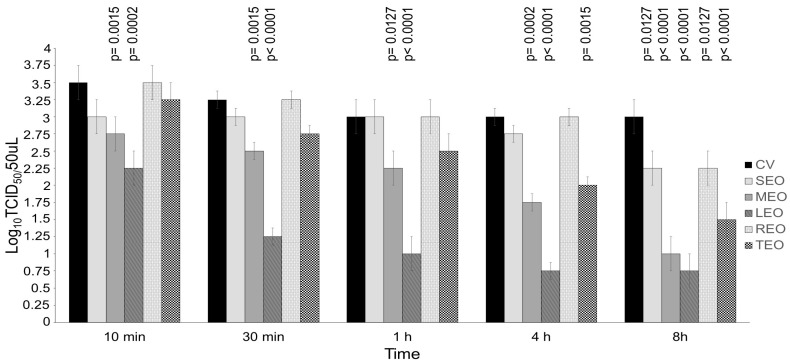
Virucidal activity of *Salvia officinalis* L. essential oil (SEO), *Melissa officinalis* L. essential oil (MEO), *Citrus lemon* essential oil (LEO), *Rosmarinus officinalis* L. essential oil and *Thymus vulgaris* L. essential oil at concentrations 100-fold over the cytotoxic threshold against the bovine viral diarrhea virus (BVDV) for 10 min, 30 min, 1 h, 4 h and 8 h at room temperature. The different virus–EO mixtures were titrated on Madin Darby bovine kidney (MDBK) cells and compared with the untreated infected cells (control virus, CV). SEO was used at 109,870.00 μg/mL, MEO was used at 12,385.70 μg/mL, LEO was used at 2986.00 μg/mL, REO was used at 34,640.00 μg/mL and TEO was used at 19,777.80 against BVDV. Viral titers of BVDV were expressed as log_10_ TCID_50_/50 μL and plotted against the EOs at different concentrations and CVs. Significant *p* values are displayed. Bars in the figures indicate the means. Error bars indicate the standard deviation.

**Table 1 antibiotics-13-00514-t001:** Chemical composition of *Salvia officinalis* L. EO (SEO). In bold are the predominant components.

N	Components	LRI	AI	*Salvia officinalis*	
				Area ± SEM	SI/MS
**1**	**α-pinene**	**930**	**931**	**9.5 ± 1**	**97**
**2**	**camphene**	**952**	**950**	**8.8 ± 1**	**97**
**3**	**β-pinene**	**982**	**981**	**8.4 ± 0.7**	**97**
4	β-myrcene	990	991	1.5 ± 0.4	96
**5**	**eucalyptol**	**1023**	**1022**	**29 ± 2**	**99**
6	γ- terpinene	1062	1064	0.60 ± 0.1	97
7	isoterpinolene	1086	1087	0.13 ± 0.04	98
**8**	**β-linalool**	**1100**	**1101**	**3.2 ± 0.8**	**96**
**9**	**camphor**	**1145**	**1146**	**23 ± 1**	**98**
**10**	**endo-borneol**	**1166**	**1167**	**2.5 ± 0.7**	**97**
11	terpinen-4-ol	1171	1171	0.2 ± 0.04	96
**12**	**α-terpineol**	**1178**	**1179**	**2 ± 0.3**	**91**
13	γ- terpineol	1195	1198	0.4 ± 0.1	94
**14**	**linalyl acetate**	**1256**	**1258**	**5 ± 0.4**	**91**
**15**	**bornyl acetate**	**1288**	**1289**	**2 ± 0.4**	**99**
16	caryophyllene	1415	1415	1.5 ± 0.8	99
17	humulene	1452	1451	0.3 ± 0.04	95
	% Characterized	/	/	98.03	/
	Others	/	/	1.97	/

LRI: Linear retention index (LRI); AI: Arithmetic index; SEM: Structural equation modeling; SI/MS: Similarity index/mass spectrum.

**Table 2 antibiotics-13-00514-t002:** Chemical composition of *Melissa officinalis* L. EO (MEO). In bold are the predominant components.

N	Components	LRI	AI	*Melissa officinalis*	
				Area ± SEM	SI/MS
1	α-pinene	930	931	0.34 ± 0.04	96
2	camphene	952	952	0.52 ± 0.05	96
3	β-thujene	968	968	0.20 ± 0.01	94
4	β-pinene	982	980	0.65 ± 0.05	91
5	eucalyptol	1023	1023	1.2 ± 0.5	98
**6**	**limonene**	**1030**	**1032**	**4.3 ± 1**	**94**
7	4-nonanone	1052	1053	0.3 ± 0.01	91
8	β-linalool	1100	1101	0.96 ± 0.25	97
9	citronellale	1168	1170	0.55 ± 0.05	93
10	α−terpineol	1178	1179	0.3 ± 0.01	80
11	citronellol	1220	1221	0.34 ± 0.04	95
**12**	**citral**	**1240**	**1240**	**43 ± 3**	**96**
**13**	**geraniol**	**1253**	**1254**	**2 ± 1**	**97**
14	α−cubebene	1347	1348	0.40 ± 0.02	98
15	eugenol	1358	1359	0.15 ± 0.01	93
16	β−ylangene	1367	1368	0.13 ± 0.01	93
17	α−copaene	1374	1375	0.9 ± 0.1	99
**18**	**geranyl aceate**	**1384**	**1385**	**2 ± 0.10**	**91**
19	β−elemene (-)	1394	1394	0.2 ± 0.01	83
**20**	**caryophyllene**	**1415**	**1415**	**25 ± 1**	**99**
21	trans-isoeugenol	1426	1427	0.3 ± 0.01	95
22	α−bergamotene	1431	1430	0.14 ± 0.01	87
**23**	**humulene**	**1452**	**1451**	**4.4 ± 0.9**	**97**
24	alloaromadendrene	1458	1458	0.2 ± 0.01	90
25	α-amorphene	1483	1484	0.9 ± 0.1	97
26	α−farnesene	1509	1508	0.13 ± 0.01	93
**27**	**caryophyllene oxyde**	**1596**	**1592**	**2.2 ± 0.9**	**91**
	% Characterized	/	/	91.71	/
	Others	/	/	8.29	/

LRI: Linear retention index (LRI); AI: Arithmetic index; SEM: Structural equation modeling; SI/MS: Similarity index/mass spectrum.

**Table 3 antibiotics-13-00514-t003:** Chemical composition of citrus lemon EO (LEO). In bold are the predominant components.

N	Components	LRI	AI	*Citrus lemon*	
				Area ± SEM	SI/MS
1	ethyl propanoate	714	714	0.1 ± 0.010	91
**2**	**α-pinene**	**930**	**931**	**2.4 ± 0.5**	**95**
**3**	**β-thujene**	**968**	**968**	**2 ± 0.2**	**86**
**4**	**β-pinene**	**982**	**980**	**14.5 ± 1**	**94**
**5**	**limonene**	**1030**	**1032**	**53 ± 5**	**93**
**6**	**γ-terpinene**	**1062**	**1064**	**5.9 ± 1**	**94**
7	terpinolene	1083	1085	0.2 ± 0.020	96
8	β-linalool	1100	1101	0.2 ± 0.020	91
9	limonene oxide, cis-	1130	1131	1 ± 0.3	96
10	limonene oxide, trans-	1138	1138	0.7 ± 0.08	91
11	α-terpineol	1178	1179	0.3 ± 0.020	80
12	cis-carveol	1222	1222	0.3 ± 0.020	96
**13**	**citral**	**1240**	**1240**	**3.8 ± 0.9**	**96**
14	Δ-carvone	1242	1242	0.15 ± 0.01	93
15	nerol acetate	1363	1364	0.8 ± 0.05	91
16	geranyl aceate	1384	1385	0.9 ± 0.06	91
17	caryophyllene	1415	1415	0.15 ± 0.01	99
18	α-bergamotene	1431	1430	0.2 ± 0.02	87
19	β-bisabolene	1504	1506	0.6 ± 0.04	95
20	caryophyllene oxyde	1596	1592	0.6 ± 0.05	91
	% Characterized	/	/	87.8	/
	Others	/	/	12.2	/

LRI: Linear retention index (LRI); AI: Arithmetic index; SEM: Structural equation modeling; SI/MS: Similarity index/mass spectrum.

**Table 4 antibiotics-13-00514-t004:** Chemical composition of *Rosmarinus officinalis* L. EO (REO). In bold are the predominant components.

N	Components	LRI	AI	*Rosmarinus officinalis* L.	
				Area ± SEM	SI/MS
1	tricyclene	920	919	0.5 ± 0.1	96
**2**	**α-pinene**	**930**	**931**	**20 ± 2**	**97**
**3**	**camphene**	**952**	**952**	**8 ± 1**	**97**
**4**	**β-pinene**	**982**	**980**	**4 ± 0.3**	**97**
**7**	**β-myrcene**	**990**	**991**	**2.2 ± 0.1**	**96**
8	α-phellandrene	1002	1003	0.34 ± 0.1	95
9	α-terpinene	1014	1014	0.4 ± 0.1	98
10	3-carene	1015	1016	0.5 ± 0.1	96
**11**	**o-cymene**	**1021**	**1021**	**3.5 ± 0.4**	**95**
**12**	**eucalyptol**	**1023**	**1022**	**23 ± 2**	**99**
13	γ- terpinene	1062	1064	0.54 ± 0.1	97
14	terpinolene	1083	1085	0.6 ± 0.1	98
15	β-linalool	1100	1101	1 ± 0.1	96
**16**	**camphor**	**1145**	**1146**	**20 ± 1**	**98**
**17**	**endo-borneol**	**1166**	**1167**	**4 ± 0.7**	**97**
18	terpinen-4-ol	1171	1171	0.8 ± 0.1	96
**19**	**α-terpineol**	**1178**	**1179**	**2.5 ± 0.3**	**96**
20	verbenone	1204	1204	1.8 ± 0.8	98
21	bornyl acetate	1288	1289	1.3 ± 0.2	98
22	ylangene	1367	1368	0.13 ± 0.02	97
**23**	**caryophyllene**	**1415**	**1415**	**2.1 ± 0.5**	**99**
24	humulene	1452	1451	0.5 ± 0.03	96
25	caryophyllene oxyde	1596	1592	0.2 ± 0.02	90
	% Characterized	/	/	97.91	/
	Others	/	/	2.09	/

LRI: Linear retention index (LRI); AI: Arithmetic index; SEM: Structural equation modeling; SI/MS: Similarity index/mass spectrum.

**Table 5 antibiotics-13-00514-t005:** Chemical composition of *Thymus vulgaris* L. EO (TEO). In bold are the predominant components.

N	Components	LRI	AI	*Thymus vulgaris*	
				Area ± SEM	SI/MS
1	propanoic acid, ethyl ester	714	714	0.10 ± 0.09	86
2	α-tricyclene	915	919	0.13 ± 0.10	94
3	α-thujene	925	926	1.2 ± 0.98	97
**4**	**α-pinene**	**930**	**931**	**2 ± 0.11**	**95**
**5**	**camphene**	**949**	**952**	**2 ± 0.70**	**96**
6	sabinene	977	977	0.1 ± 0.3	93
7	β-pinene	978	978	0.6 ± 0.03	94
8	β-myrcene	985	991	1.4 ± 0.25	86
9	α-phellandrene	1001	1003	0.15 ± 0.01	91
**10**	**p-cymene**	**1024**	**1024**	**20 ± 2.50**	**95**
11	limonene	1033	1027	0.6 ± 0.01	91
12	eucalyptol	1023	1031	0.9 ± 0.07	99
13	cis-β-terpineol	1145	1147	0.13 ± 0.01	90
**14**	**γ-terpinene**	**1063**	**1059**	**9 ± 0.9**	**94**
15	α-terpinolene	1085	1089	0.12 ± 0.01	81
**16**	**β-linalool**	**1097**	**1098**	**4 ± 1**	**97**
17	camphor	1145	1146	1.7 ± 0.8	98
18	borneol	1166	1167	1.9 ± 0.9	97
19	terpinen-4-ol	1172	1174	1.8 ± 0.9	96
20	α-terpineol	1189	1190	0.12 ± 0.01	86
21	methyl thymol, ether	1235	1235	0.4 ± 0.2	90
22	isothymol methyl ether	NA	1244	0.4 ± 0.2	94
**23**	**thymol**	**1290**	**1290**	**47 ± 2**	**94**
**24**	**caryophyllene**	**1417**	**1418**	**2 ± 0.9**	**99**
25	caryophyllene oxide	1581	1592	0.6 ± 0.1	91
	% Characterized	/	/	98.35	/
	Others	/	/	1.65	/

LRI: Linear retention index (LRI); AI: Arithmetic index; SEM: Structural equation modeling; SI/MS: Similarity index/mass spectrum.

## Data Availability

The data that support the findings of this study are described in the manuscript.

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
