# Peer review of "In Vitro Virucidal Activity of Different Essential Oils against Bovine Viral Diarrhea Virus Used as Surrogate of Human Hepatitis C Virus"

_antibiotics, 2024, doi:10.3390/antibiotics13060514_

Round 1

Reviewer 1 Report

Comments and Suggestions for Authors

Due the risk of hepatocellular carcinoma increment in humans caused by Hepatitis C virus the antiviral and the virucidal actvitivies of Salvia officinalis, Melissa officinalis, Citrus lemon, Rosmarinus officinalis and Thymus vulgaris essential oils were studied using Bovine viral diarrhea virus (BVDV) as a surrogate of Hepatitis C virus. The characterization of these essential oils was correctly determined by gas chromatography technique finding that 25 chemical compounds accounts the 98% of the chemical composition finding compounds as thymol, p-cymene, γ-terpinene, β-linalool, α-pinene, camphene and caryophyllene. Cytotoxicity of all essential oils was determined by microscopic examination of cell morphology and measurement of cell viability by the XTT colorimetric method after exposing the cells to various concentrations of the compound for 72 hours. It was found that Melissa officinalis and Citrus lemon essential oils were able to decrease considerably BVDV infectivity. Further investigations are required to improve the performance of both essential oils, to investigate the mode of action on BVDV, and to determine if other BVDV strains might display different susceptibility to Melissa officinalis and Thymus vulgaris essential oils.  The research is very well described and developed.

Author Response

Rebuttal

Dear Reviewer #1,

herein you can find a point-by-point response to your comments for the manuscript antibiotics-3004565 entitled “In vitro virucidal activity of different essential oils against Bovine

Viral Diarrhea Virus used as surrogate of human hepatitis C virus” submitted to Antibiotics.

Thanks in advance

Best regards

Michele Camero

Reviewer #1

Due the risk of hepatocellular carcinoma increments in humans caused by Hepatitis C virus the antiviral and the virucidal activities of Salvia officinalis, Melissa officinalis, Citrus lemon, Rosmarinus officinalis and Thymus vulgaris essential oils were studied using Bovine viral diarrhea virus (BVDV) as a surrogate of Hepatitis C virus. The characterization of these essential oils was correctly determined by gas chromatography technique finding that 25 chemical compounds account the 98% of the chemical composition finding compounds as thymol, p-cymene, γ-terpinene, β-linalool, α-pinene, camphene and caryophyllene. Cytotoxicity of all essential oils was determined by microscopic examination of cell morphology and measurement of cell viability by the XTT colorimetric method after exposing the cells to various concentrations of the compound for 72 hours. It was found that Melissa officinalis and Citrus lemon essential oils were able to decrease considerably BVDV infectivity. Further investigations are required to improve the performance of both essential oils, to investigate the mode of action on BVDV, and to determine if other BVDV strains might display different susceptibility to Melissa officinalis and Thymus vulgaris essential oils.  The research is very well described and developed.

General reply to R1: We thank the referee for his/her appreciation for the manuscript. The efforts of the co-authors are surely paid back by these considerations.

Reviewer 2 Report

Comments and Suggestions for Authors

After a thorough review of the manuscript entitled “In vitro virucidal activity of different essential oils against Bovine Viral Diarrhea Virus used as surrogate of human hepatitis C virus”, I highlighted some points that can be taken into consideration by the authors to improve the work.

1. Lines 23-24: Include the acronym for the essential oil of all species.

2. Lines 41-42: Remove the link from the introduction and add it to the reference list.

3. Lines 85-86: The species Phyllanthus amarus, Eclipta alba and Acacia nilotica mentioned in the introduction were not used in the virucidal tests carried out in this study. I suggest deleting this information and including comments about the antiviral/virucidal potential of Salvia officinalis, Melissa officinalis, Citrus lemon, Rosmarinus officinalis, and Thymus vulgaris.

4. The authors must improve the justification related to the development of this study based on the following questions: Why investigate the virucidal potential of essential oils from these five different species? What is the main novelty of this study? Please make this information clear in the introduction of the manuscript.

5. Line 94: It is important to mention and comment on each specific table throughout the results text.

6. Lines 138-140: I suggest including photos showing the morphology of cells after exposure to different essential oils and comparing them with photos of the control.

7. Lines 148-150: Cytotoxicity results must be presented in a table or graph. Include the standard deviation of the data obtained.

8. Figures 1 and 2 were not cited in the text of the results. Please mention each figure in the virucidal activity section.

9. Lines 219-241: The first three paragraphs of the discussion are very broad and do not discuss the specific results presented in this manuscript. I suggest deleting.

10. Line 341: Which parts of each species were used to obtain essential oils? Even if these products were purchased commercially, this information must be included in the manuscript.

11. Lines 384-389: Why each essential oil was evaluated in different concentrations? Wouldn't it be more appropriate to standardize a single series of concentrations and apply it to all products tested in the cytotoxicity assay?

12. Line 399: I suggest specifying the concentrations of essential oils used in virucidal activity.

13. Line 396-402: Was a positive control not used? In this test, it is important to use a commercial virucide to compare the results with essential oils.

Author Response

Rebuttal

Dear Reviewer #2,

herein you can find a point-by-point response to your comments for the manuscript antibiotics-3004565 entitled “In vitro virucidal activity of different essential oils against Bovine

Viral Diarrhea Virus used as surrogate of human hepatitis C virus” submitted to Antibiotics.

Thanks in advance

Best regards

Michele Camero

Reviewer #2

After a thorough review of the manuscript entitled “In vitro virucidal activity of different essential oils against Bovine Viral Diarrhea Virus used as surrogate of human hepatitis C virus”, I highlighted some points that can be taken into consideration by the authors to improve the work.

R2.1. Lines 23-24: Include the acronym for the essential oil of all species.

Reply to R2.1. This was done.

R2.2. Lines 41-42: Remove the link from the introduction and add it to the reference list.

Reply to R2.2. The link was removed, and a reference was added in the bibliography, as suggested.

R2.3. Lines 85-86: The species Phyllanthus amarus, Eclipta alba and Acacia nilotica mentioned in the introduction were not used in the virucidal tests carried out in this study. I suggest deleting this information and including comments about the antiviral/virucidal potential of Salvia officinalis, Melissa officinalis, Citrus lemon, Rosmarinus officinalis, and Thymus vulgaris.

Reply to R2.3 We removed the sentence as suggested and we replaced it with information regarding the antiviral/virucidal potential of Salvia officinalis, Melissa officinalis, Citrus lemon, Rosmarinus officinalis, and Thymus vulgaris essential oils, as suggested (see lines 87-96, page 2)

R2.4. The authors must improve the justification related to the development of this study based on the following questions: Why investigate the virucidal potential of essential oils from these five different species? What is the main novelty of this study? Please make this information clear in the introduction of the manuscript.

Reply to R2.4. The motivation of the use of the five essential oils against BVDV and the novelty of the study have been have been implemented in the introduction (see lines 97-103, page 2-3).

R2.5. Line 94: It is important to mention and comment on each specific table throughout the results text.

Reply to R2.5. All the tables have been properly commented in the results providing the most significant information on the composition of essential oils.

R2.6. Lines 138-140: I suggest including photos showing the morphology of cells after exposure to different essential oils and comparing them with photos of the control.

Reply to R2.6. Figure 1 was added and commented in the text (see lines 153-160, page 6), as suggested. The novel figure contains microscopy images of cytotoxic effects of the essential oils on MDBK cells compared with untreated (control) MDBK cells. Numbering of further figures changed accordingly.

R2.7. Lines 148-150: Cytotoxicity results must be presented in a table or graph. Include the standard deviation of the data obtained.

Reply to R2.7. Cytotoxicity results and their standard deviation have been provided in Figure 2 and commented in the text (lines 163-191 page 7-8, as requested).

R2.8. Figures 1 and 2 were not cited in the text of the results. Please mention each figure in the virucidal activity section.

Reply to R2.8. Due to the referee comment R2.6 and R2.7 the numbering of figures changed to figure 3 and 4 that were cited in the text, as requested.

R2.9. Lines 219-241: The first three paragraphs of the discussion are very broad and do not discuss the specific results presented in this manuscript. I suggest deleting.

Reply to R2.9. The first three paragraphs have been removed, as suggested.

R2.10. Line 341: Which parts of each species were used to obtain essential oils? Even if these products were purchased commercially, this information must be included in the manuscript.

Reply to R2.10. We have added the requested information in the text (lines 376-377 page 12).

R2.11. Lines 384-389: Why each essential oil was evaluated in different concentrations? Wouldn't it be more appropriate to standardize a single series of concentrations and apply it to all products tested in the cytotoxicity assay?

Reply to R2.11. We thank the referee R2 for the suggestion and we will perform the experiments using standardized concentrations in future studies. However, in this study we followed a procedure previously described (doi: 10.1016/j.heliyon.2024.e30492) for the assessment of the concentration at which the viability of treated cells decreased to 20% compared to control cells regarded as CC20.

R2.12. Line 399: I suggest specifying the concentrations of essential oils used in virucidal activity.

Reply to R2.12. We added this information, as suggested (see lines 419-422, page 13).

R2.13. Line 396-402: Was a positive control not used? In this test, it is important to use a commercial virucide to compare the results with essential oils.

Reply to R2.13. Our study did not include a positive control, following other previous studies in the literature regarding the in vitro evaluation of virucidal/antiviral activity of EOs against viruses (Pellegrini et al., 2023 https://doi.org/10.3390/antibiotics12020322; Cozzi et al., 2023 https://doi.org/10.3390/v15030682; Kim et al., 2017 https://doi.org/10.4315/0362-028X.JFP-16-162).

However, due to the moderate activity exhibited by SEO against BVDV in a previous study (https://doi.org/10.3390/pathogens10040403), SEO could be regarded as a “positive control” for our study. Nevertheless, the results obtain in the previous study differed in terms of chemical composition of SEO that depends on harvest time of the plant, method of preparation, storage conditions, expiration of the compounds. Moreover, some features of the plant i.e., (age of the plant, meteorological conditions during their growth, soil composition, light, temperature and level of rainfall) could also affect the parallelism with our study. Also, methods used in our study for SEO differed from those applied in the previous study (https://doi.org/10.3390/pathogens10040403).

Reviewer 3 Report

Comments and Suggestions for Authors

This study presents the virucidal activity of five EOs, Salvia officinalis L. EO, Melissa officinalis L. EO (MEO), Citrus lemon EO (LEO), Rosmarinus officinalis L. EO and Thymus vulgaris L. EO against BVDV was evaluated.The review ideas appear justified. The manuscript is also well organized. Listed are some comments regarding the submitted manuscript.

1.      Line 18-32: Why’s abstract section divided into three paragraphs? Please correct it following the journal format.

2.      Line 39-40: What’s the difference between four distinct genera of Flaviviruses? Please add this information in more detail.

3.      Line 90: What’s the source of Essential oils (EOs)? Please mention this point in the text.

4.      Line 218, Discussion section: Even Table 1-5 showed the chemical composition of the tested Eos, the discussion section has not shown the comparison and explain why it have different in chemical composition?

5.      Line 396: Please provide the reference of “Virucidal activity assay”.

Author Response

Rebuttal

Dear Reviewer #3,

herein you can find a point-by-point response to your comments for the manuscript antibiotics-3004565 entitled “In vitro virucidal activity of different essential oils against Bovine

Viral Diarrhea Virus used as surrogate of human hepatitis C virus” submitted to Antibiotics.

Thanks in advance

Best regards

Michele Camero

Reviewer #3

This study presents the virucidal activity of five EOs, Salvia officinalis L. EO, Melissa officinalis L. EO (MEO), Citrus lemon EO (LEO), Rosmarinus officinalis L. EO and Thymus vulgaris L. EO against BVDV was evaluated. The review ideas appear justified. The manuscript is also well organized. Listed are some comments regarding the submitted manuscript.

R3.1.  Line 18-32: Why’s abstract section divided into three paragraphs? Please correct it following the journal format.

Reply to R3.1. This was corrected.

R3.2. Line 39-40: What’s the difference between four distinct genera of Flaviviruses? Please add this information in more detail.

Reply to R3.2.  Further information regarding difference of Flavivirus genera have been added in the text, as requested (see lines 40-47, pages 1-2).

R3.3. Line 90: What’s the source of Essential oils (EOs)? Please mention this point in the text.

Reply to R3.3. This information was added in the text in the Materials and Methods section(lines 376-377, page 12).

R3.4. Line 218, Discussion section: Even Table 1-5 showed the chemical composition of the tested Eos, the discussion section has not shown the comparison and explain why it have different in chemical composition?

Reply to R3.4.  In the discussion we focused on the chemical composition of MEO and LEO because they represent the compounds with the most significant virucidal action in our study. We commented the efficacy of some chemical compound of the mixtures comparing the results with those obtained in the literature (see lines 329-342, page 11).

 3.5. Line 396: Please provide the reference of “Virucidal activity assay”.

Reply to R3.5.  We added the reference as requested (see lines 423-424, page 13).

Round 2

Reviewer 2 Report

Comments and Suggestions for Authors

The authors responded to all my comments and the manuscript can be accepted after minor corrections. The scientific name of the species Rosmarinus officinalis (Line 133) and Thymus vulgaris (Line 142) must be in italics. The quality, size, and resolution of Figure 2 must be improved to facilitate the visualization of the information.

Author Response

Dear reviewer,

herein you can find a point-by-point response to the reviewers' comments for the manuscript antibiotics-3004565 entitled “In vitro virucidal activity of different essential oils against Bovine

Viral Diarrhea Virus used as surrogate of human hepatitis C virus” submitted to Antibiotics.

Thanks in advance

Best regards

Michele Camero

Reviewer #2

The authors responded to all my comments and the manuscript can be accepted after minor corrections.

R2.1 The scientific name of the species Rosmarinus officinalis (Line 133) and Thymus vulgaris (Line 142) must be in italics.

Reply to R2.1 All the scientific names have been checked and put in italics throughout the manuscript, as suggested.

R2.2 The quality, size, and resolution of Figure 2 must be improved to facilitate the visualization of the information.

Reply to R2.2 Figure 2 was improved, as requested.

Reviewer 3 Report

Comments and Suggestions for Authors

The authors have been revised at the suggestion of reviewers.

I agreed to the revised manuscript for publication. 

Author Response

We thank the reviewer for the appreciation of the revision performed and for the precious suggestions provided from him/her.